# Reducing cardiovascular disease risk among families with familial hypercholesterolaemia by improving diet and physical activity: a randomised controlled feasibility trial

Fiona Jane Kinnear ![ORCID],[1,2] Fiona E Lithander,[1,3] Aidan Searle ![ORCID],[1,2] Graham Bayly,[4] Christina Wei,[5] David J Stensel ![ORCID],[6,7] Alice E Thackray,[6,7] Linda Hunt,[1,2] Julian P H Shield[1,2]

For numbered affiliations see end of article.

**Correspondence to**
Fiona Jane Kinnear;
fiona.kinnear@bristol.ac.uk

## ABSTRACT

**Objective** Familial hypercholesterolaemia (FH) elevates low-density lipoprotein cholesterol (LDL-C) and increases cardiovascular disease (CVD) risk. This study aimed to provide evidence for the feasibility of conducting a randomised controlled trial to evaluate the efficacy of an intervention designed to improve diet and physical activity in families with FH.

**Design** A parallel, randomised, waitlist-controlled, feasibility pilot trial.

**Setting** Three outpatient lipid clinics in the UK.

**Participants** Families that comprised children (aged 10–18 years) and their parent with genetically diagnosed FH.

**Intervention** Families were randomised to either 12-week usual care or intervention. The behavioural change intervention aimed to improve dietary, physical activity and sedentary behaviours. It was delivered to families by dietitians initially via a single face-to-face session and then by four telephone or email follow-up sessions.

**Outcome measures** Feasibility was assessed via measures related to recruitment, retention and intervention fidelity. Postintervention qualitative interviews were conducted to explore intervention acceptability. Behavioural (dietary intake, physical activity and sedentary time) and clinical (blood pressure, body composition and blood lipids) outcomes were collected at baseline and endpoint assessments to evaluate the intervention's potential benefit.

**Results** Twenty-one families (38% of those approached) were recruited which comprised 22 children and 17 adults with FH, and 97% of families completed the study. The intervention was implemented with high fidelity and the qualitative data revealed it was well accepted. Between-group differences at the endpoint assessment were indicative of the intervention's potential for improving diet in children and adults. Evidence for potential benefits on physical activity and sedentary behaviours was less apparent. However, the intervention was associated with improvements in several CVD risk factors including LDL-C, with a within-group mean decrease of 8% (children) and 10% (adults).

### Strengths and limitations of this study

- ► This is the first study to assess the feasibility and acceptability of delivering a behaviour change intervention, which aims to enable attainment of the dietary and physical activity treatment recommendations, to families affected by familial hypercholesterolaemia (FH).
- ► The quantitative and qualitative measures employed in this multicentred randomised controlled trial have provided evidence of the feasibility and acceptability of conducting a future, adequately powered trial, subject to identified refinements.
- ► An adequately powered randomised controlled trial is required to formally evaluate the effectiveness of the intervention on behavioural and clinical outcomes.
- ► The sample comprised mostly highly educated white European participants and the findings may not be generalisable to all individuals living with FH.

**Conclusions** The study's recruitment, retention, acceptability and potential efficacy support the development of a definitive trial, subject to identified refinements.

**Trial registration number** ISRCTN24880714.

## INTRODUCTION

Familial hypercholesterolaemia (FH) is a genetic disease characterised by elevated levels of low-density lipoprotein cholesterol (LDL-C) from birth.[1] The disease confers an 18-fold increased risk of cardiovascular disease (CVD), accounting for 1 in every 17 CVD cases worldwide.[2] The heterozygous genotype of FH affects 1 in 311 individuals globally. Pharmacological treatment significantly reduces LDL-C, and thus CVD risk, especially when initiated in childhood.[3]

However, CVD risk remains elevated and its onset and severity vary substantially.[4 5]

Environmental factors, including lifestyle behaviours, may explain some of the variance in CVD risk,[6] mediated via LDL-C reductions additive to that attained by pharmacological treatment or effects on other CVD risk factors.[7 8] Both pathways are important as LDL-C treatment goals are often not achieved,[9] and hypertension, type 2 diabetes mellitus and obesity are independently associated with CVD risk among adults with FH.[7] Accordingly, adjuvant to pharmacological therapy, clinical guidelines recommend individuals with FH should be encouraged to be physically active, maintain a healthy weight and consume a healthy diet.[10] There is no evidence from randomised controlled trials (RCTs) demonstrating the effectiveness of these recommendations on CVD risk in individuals with FH.[11 12]

A previous RCT sought to address this evidence gap,[13] but the intervention was implemented with poor fidelity and failed to significantly improve dietary or physical activity behaviours.[14] Less than half of children and adults with FH adhere to dietary and physical activity treatment recommendations,[15 16] with several factors identified as barriers including the asymptomatic nature prior to onset of CVD and the perception that it is unnecessary when also taking medication.[17] In order to explore the true effects of diet and physical activity on CVD risk, it is essential that interventions can successfully change these behaviours. Given the high prevalence of FH, any intervention must also be feasible to implement within the time and resource constraints of health services.[10] Therefore, we previously developed a pragmatic behaviour change intervention specifically designed to address the determinants of dietary and physical activity behaviours among individuals with FH.[18] As the inheritance pattern of FH means all affected children will have one affected parent, it was delivered to families.

According to the Medical Research Council (UK) guidance,[19] healthcare interventions should undergo feasibility testing prior to evaluation in a full-scale RCT. Therefore, this trial aimed to provide evidence for the feasibility and acceptability of conducting a future, adequately powered RCT and identify any refinements to the RCT design or intervention required. The six objectives were to:

1. Assess the feasibility of recruiting, randomising and retaining families with FH.
2. Evaluate the feasibility of collecting valid outcome measures.
3. Assess the feasibility of implementing the intervention with sufficient fidelity.
4. Explore the participant acceptability of the intervention.
5. Explore the potential effectiveness of the intervention on behavioural and clinical outcomes.
6. Estimate the sample size required to adequately power a future RCT.

## METHODS
The study protocol is published,[20] and a Consolidated Standards of Reporting Trials checklist[21] is displayed in online supplemental file 1.

### Trial design and setting
The study was a parallel, two-arm, randomised, controlled, feasibility pilot trial[22] comparing a diet and physical activity intervention against usual care waitlist control among families living with FH. It was conducted across three National Health Service (NHS) Foundation Trust sites in England, UK.

### Participants
The study aimed to recruit families that comprised child aged 10–18 years and their parent, both with a genetic diagnosis of FH and established on their current treatment (lifestyle and/or pharmacological) regimen ≥1 month. Families could comprise multiple affected children and non-affected parents or carers in instances in which affected parents were unwilling, unable or ineligible to participate. If this was not possible, children could participate without parental involvement.

### Recruitment and randomisation
All eligible children receiving care from outpatient lipid clinics at the study sites were invited to participate, along with their parent, via invitation letters and/or when attending routine clinic appointments. Written informed consent (or assent and parental consent for children under 16 years old) was obtained from all participants. After baseline data collection, families were randomised to receive usual care and waitlist control or the intervention on a 1:1 basis, stratified by study site. This was carried out by a database manager using pre-prepared, password-protected, randomised lists. It was not possible to blind participants or research staff to the randomisation.

### Sample size
As this was a feasibility study, a formal sample size calculation was not required.[21] The recruitment of 24 families, based on the local number of eligible families, was deemed to be large enough to address the study objectives.

### Intervention
The development and content of the behaviour change intervention are described in detail elsewhere[18] and in online supplemental file 2. In brief, the intervention aimed to enable individuals to reduce dietary intakes of total fat, saturated fat and cholesterol; increase intakes of unsaturated fats, fibre, fruits, vegetables and plant stanol or sterol-fortified foods; and reduce sedentary behaviour and increase physical activity. The behaviour change wheel, which brings together 19 different theories of behaviour change,[23] was applied to the findings of a qualitative evidence synthesis to identify the determinants of treatment behaviours.[17] Twenty-six behavioural change techniques (BCTs) were then incorporated into the intervention to target these.[18] The intervention was

delivered to families by one of two dietitians as an initial 1-hour face-to-face session and four email or telephone follow-up sessions over a 12-week intervention period. The dietitians received comprehensive manuals which detailed the content and BCTs to be delivered to families at each session (online supplemental file 2).

### Comparator (waitlist, usual care control)
Families were informed that they were on a waitlist to receive the intervention at the end of the 12-week study period and received usual care, which comprised an annual outpatient lipid clinic appointment. Adults, but not children, in both groups had previously received dietetic counselling on FH diagnosis as part of usual care. No participant received any further dietetic advice during the trial control period.

### Pharmacological treatment
Participants in both groups were instructed to continue with their prescribed pharmacological treatment, if applicable. Any self-reported changes to medication were documented and validated through confirmation from medical records.

### Outcome measures
#### Feasibility outcomes (study objectives 1–3)
Recruitment, randomisation and retention rates were collected for families, as the study aimed to recruit children and their parent. Checklists completed by dietitians after each intervention session were analysed to produce outcomes related to the *dose* (number and duration of sessions delivered) and *fidelity* (inclusion of prespecified content sections and BCTs) of the intervention delivered to each family.[24] Dietitians documented any adaptations made to the intervention. The proportion of participants from whom valid outcome measures were collected was used to assess the feasibility of obtaining these measures. While outcome measures were collected from all adults, data are only analysed and presented for adults affected with FH.

#### Acceptability outcomes (study objective 4)
After study completion, 10 families took part in face-to-face or telephone qualitative interviews to explore the acceptability of the intervention. Purposive sampling was employed to yield maximum variation within this subsample. The sample included four families from the control group who were interviewed after they received the intervention on study completion. All interviews were conducted by FJK and were audio recorded and transcribed *verbatim*.

#### Behavioural and clinical outcomes (study objectives 5 and 6)
It was not appropriate to test the effectiveness hypothesis, as the study was not sufficiently powered to do so.[25] However, all outcome measures to be assessed in a future trial were measured and the potential effectiveness of the intervention on behavioural (diet, physical activity and sedentary behaviours) and clinical (selected CVD risk factors and quality of life (QoL)) outcomes was explored. Clinical outcomes were measured by the research team at baseline and endpoint assessment visits before and after the 12-week intervention or control period. Behavioural outcomes were measured in the week preceding these visits. A summary is provided below, with full details elsewhere[20] and in online supplemental file 3.

Participants recorded their dietary intake on four non-consecutive days (including one weekend day) using Intake24, a validated online 24-hour recall tool.[26 27] At least 3 days were required for analysis and data were analysed to produce outcome measures related to the intervention dietary targets. Participants wore two accelerometers (ActiGraph GT3X+ and activPAL3) for seven consecutive days to measure moderate and vigorous physical activity (MVPA) and time spent being sedentary during waking hours. At least four valid days (including one weekend day) were required for analysis.

Height was measured using a stadiometer and weight (kg), body fat (%) and fat-free mass (kg) using bioelectrical impedance scales (Tanita MC-780MA). At sites where this equipment was unavailable, weight was measured using medical scales. Body mass index (BMI) was determined from the height and weight measurements and systolic and diastolic blood pressures were measured using automated oscillometric devices (Dinamap, V100). Health-related quality of life (HRQoL) was measured in children using a Pediatric Quality of Life Inventory and in adults using the EuroQol Group EQ-5D-3L health questionnaires.[28 29] Fasted blood samples were collected to measure total cholesterol, triglyceride and high-density lipoprotein cholesterol using standard methodology and reagents on local NHS laboratory autoanalysers (details in online supplemental file 3). LDL-C was estimated using the Friedewald equation.[30] The results of lipidomic analyses will be reported in the future due to the closure of the relevant metabolomics laboratory because of the coronavirus.

### Data analysis
Feasibility outcomes were analysed descriptively and the qualitative outcomes analysed using the six stages of thematic analysis described by Braun and Clarke.[31] The analysis of the behavioural and clinical outcomes was carried out separately for children and adults. For parametric outcomes, the baseline and endpoint measures for intervention and control groups were summarised with means and SDs. The endpoint means in the intervention and control groups were compared with and without adjustment for the baseline values using regression analysis,[32] and their differences reported with 95% CIs. For non-parametric outcomes, baseline and endpoint measures and within-group pre-to-post changes for intervention and control groups were summarised by median and range. To maximise the power of the models, all valid data collected at baseline and endpoint assessments were included in the analyses.

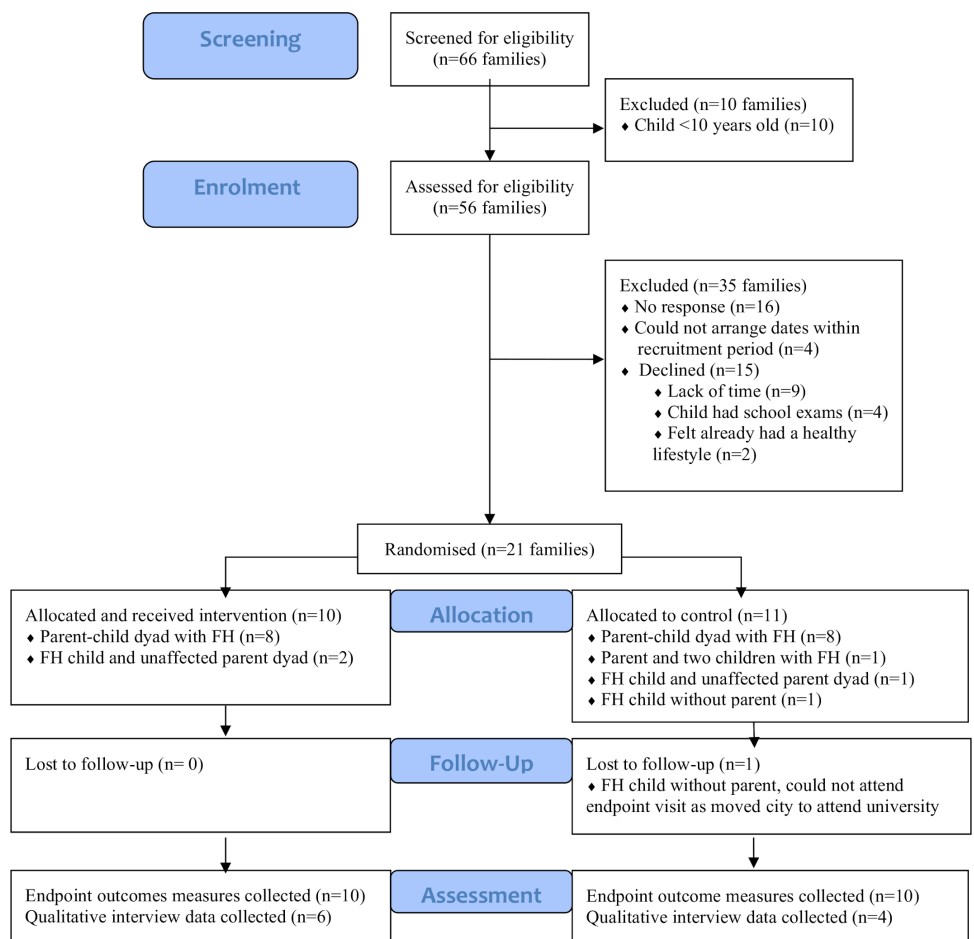

**Figure 1** Consolidated Standards of Reporting Trials (CONSORT) diagram of recruitment of families across the three sites. FH, familial hypercholesterolaemia.

## Patient involvement

As detailed in the protocol,[20] patients with FH under the care of the University Hospitals Bristol NHS Foundation Trust participated in the development work underpinning this research. They gave feedback about the initial design of the intervention and data collection methods.

## RESULTS

### Recruitment, randomisation and retention (study objective 1)

Recruitment took place between August 2018 and December 2019 and ceased due to study timelines. Recruitment to the trial was 38% of eligible families, with 21 of the target 24 families recruited, 18 (86%) of which comprised a child and their parent with FH (figure 1). All families were successfully randomised and retention was high at 97%. The baseline characteristics of the participants (with FH) are displayed in table 1.

### Collection of clinical and behavioural outcome measures (study objective 2)

All families who completed the study attended all research visits, although one child was unable to accompany his/her parent to the endpoint assessment. Clinical outcome measures were collected from all other participants, with

the exception of body composition, due to unavailability of the required equipment at two of the three sites, and blood lipid profile, due to needle phobia (n=2; children), nurse unavailability (n=3; 2 child, 1 adult) and unsuccessful venepuncture (n=2; 1 child, 1 adult). All adults selected the maximum values in the HRQoL questionnaires and due to this ceiling effect, the data were not analysed. Valid paired (baseline and endpoint) dietary outcome measures were obtained from >80% of participants, with missing data due to participants (n=5; 4 child, 1 adult) not completing the required minimum 3 days of dietary recalls. Due to battery malfunctions (n=12; 6 child, 6 adult) and non-wear due to skin reactions (n=3; children), paired sedentary outcome measures were only collected from 57% and 65% of children and adults, respectively.

### Intervention implementation and fidelity outcomes (study objective 3)

All families received the initial intervention session and the duration averaged 66 min, 10% higher than the intended 60 min. All families received the first three follow-up sessions, only 60% of families opted to receive the fourth follow-up session. Two families chose to receive

**Table 1** Baseline characteristics of the study participants*

| Characteristic | Adults | | Children | |
|---|---|---|---|---|
| | Control (n=9) | Intervention (n=8) | Control (n=12) | Intervention (n=10) |
| Gender female, n (%) | 4 (44) | 4 (50) | 6 (50) | 5 (50) |
| Age (years), mean (SD); range | 51 (5); 43–59 | 42 (6); 32–49 | 15 (3); 10–18 | 13 (2); 10–16 |
| Ethnicity, n (%) | | | | |
| White European | 8 (89) | 7 (88) | 9 (75) | 9 (90) |
| Asian | 1 (11) | 1 (12) | 3 (25) | 1 (10) |
| Managerial or professional job, n (%) | 7 (78) | 8 (100) | – | – |
| Treatment, n (%) | | | | |
| Lifestyle only | 0 | 1 (13) | 2 (17) | 3 (30) |
| Statin medication | 2 (22) | 2 (25) | 10 (80) | 7 (70) |
| Statin and ezetimibe medication | 6 (67) | 5 (63) | 0 | 0 |
| PCSK9 inhibitor injections | 1 (11) | 0 | 0 | 0 |
| FH pathogenic variant location, n (%) | | | | |
| LDLR | 7 (78) | 5 (63) | 10 (83) | 7 (70) |
| APOB | 2 (12) | 2 (25) | 2 (17) | 2 (20) |
| PCSK9 | 0 | 1 (12) | 0 | 1 (10) |
| Risk factors, n (%) | | | | |
| Current smoker | 0 | 1 (13) | 0 | 0 |
| Previous smoker | 4 (44) | 1 (13) | 0 | 0 |
| Hypertension | 2 (22) | 1 (13) | 0 | 0 |
| Overweight weight status† | 4 (44) | 4 (50) | 2 (17) | 2 (20) |
| Obese weight status‡ | 0 | 0 | 1 (8) | 1 (10) |

*Only adults with FH included in this table.
†Overweight in adults defined as BMI≥25–29.9 kg/m$^2$ and BMI centile ≥ 91st percentile for children.
‡Obese defined as BMI≥30 kg/m$^2$ for adults and BMI centile ≥ 98th percentile for children.
APOB, apolipoprotein B; BMI, body mass index; FH, familial hypercholesterolaemia; LDLR, low-density lipoprotein receptor; PCSK9, proprotein convertase subtilisin/kexin type 9.

follow-up sessions via email and eight opted for telephone with an average duration of 25 min, which was below the intended 30 min. The fidelity to the content and BCTs intended to be delivered in each session was high, with only one content section ('*barriers and solutions*', detailed in online supplemental file 2) not delivered to 20% of families, due to the initial session exceeding the intended 60 min and the family having to leave. In response to the individual needs of the families, four BCTs were not delivered to all families and 80% of families received additional BCTs during the follow-up sessions. Two further BCTs were not delivered to 20% of families, as the required information (dietary intake and body composition data) was unavailable for the initial session. Online supplemental file 2 provides details of these omitted and additional BCTs.

## Acceptability results (study objective 4)

Overall, the participants were positive towards the content of the intervention, which was perceived to have provided the knowledge and motivation to change behaviours: '*If no-one's telling you what you have to change, then it's harder to think about it*' (Parent). The follow-up sessions were valued as they provided additional information and/or motivational support: '*If we'd not had any contact, it might have been difficult to keep on track*' (Parent). The stepwise approach to goal setting and the family-based delivery of the intervention facilitated behaviour change: '*it's just easier if you have someone else doing it with you*' (Child). If suitable alternatives could be identified, the dietary goals were perceived to be easy to implement and participants intended to continue with the swaps: '*I got used to the cheese and the brown pasta. It just tastes better now than the other pasta*' (Child). Having their child enjoy these foods increased acceptability: '*To see her eating broccoli and green beans, it's what every mum wants*' (Adult). For those who were inactive previously, changing physical activity behaviours proved harder to implement, requiring '*discipline*' (Adult) and if they did not find an activity they enjoyed, many continued to struggle. The acceptability of changing behaviours was harder when routines changed during school holidays. However, participants viewed these as short-term lapses and returned to their new habits after: '*The summer holidays have been tricky because you're always out of a routine. When we're back at school it will be easier*' (Adult).

### Behavioural and clinical outcomes (study objectives 5 and 6)

The baseline data collected from the withdrawn participant were included in the analysis. The results suggest that the intervention may be effective in decreasing intake of total fat, saturated fat and cholesterol, and increasing intake of fibre, fruits and vegetables (table 2). No participants in either group consumed plant stanol or sterol-fortified products at baseline. This remained true for the control group, whereas all but one participant in the intervention group reported consuming a yoghurt containing 2 g of plant stanols or sterols every day throughout the intervention. The within-group median change between baseline and endpoint assessment is indicative of a positive effect of the intervention on increasing daily MVPA and decreasing daily sedentary time in adults, but not children (table 2). However, the data suggest MVPA also increased among adults in the control group, meaning it is not possible to attribute the improvements solely to the intervention. The direction of trends suggests a favourable effect of the intervention on diastolic blood pressure, total cholesterol and LDL-C in adults and children (table 2). Additionally, the within-group median change suggests that the intervention may be associated with improvements in BMI, body composition and HRQoL in children (table 3).

The within-group mean decrease in LDL-C for children in the intervention group was 0.29 (0.58) mmol/L and for adults was 0.33 (0.77) mmol/L, which is an approximate decrease of 8% and 10% from baseline, respectively. An LDL-C reduction of 10% was deemed to be of clinical significance and the sample size calculation was based on this. Using the mean baseline LDL-C and SDs in children in the intervention group, and a 5% level of significance, a sample size of 42 children in each group was calculated to have 80% statistical power to show a difference between a 10% decrease in LDL-C and no difference in the control group. Using the same methods, a sample size of 68 adults in each group was calculated. To account for non-collection of behavioural outcomes and participation of families with non-affected parents, a future trial should aim to recruit 160 families. Using the 38% recruitment rate, this would require a potential recruitment pool of 420 eligible families.

If multiple affected children from a family were to be recruited to any following trial, an estimate of the intraclass correlation would be needed to calculate an inflation factor for the sample size. Furthermore, mixed models may be needed in the final analyses. As only one family in the present trial included more than one affected child, it was not possible to estimate the intraclass correlation.

## DISCUSSION

This trial has demonstrated that it is feasible to recruit, randomise and retain families with FH into a 12-week trial, and to implement and evaluate a dietary and physical activity intervention, which is acceptable to families. While this study was not sufficiently powered, the potentially favourable effects on CVD risk factors are encouraging and warrant further investigation in a full-scale RCT, subject to the identified refinements.

The 8%–10% reduction in LDL-C that was observed in this trial is clinically significant. It is in fact comparable to that achieved through the addition of further medication, such as one that inhibits the intestinal absorption of cholesterol, to standard statin therapy.[33] Many children and adults with FH do not meet LDL-C treatment targets, even when using maximally licensed doses of medication.[3] There is no LDL-C concentration below which the beneficial effects on CVD risk cease to improve,[34] and 'the lower, the better' approach is advocated.[10] Any additional LDL-C reduction that can be achieved through non-pharmacological approaches is critically important, highlighting the potential benefit of the LDL-C finding.

Most families in this trial were recruited at paediatric lipid clinics, held every 3–6 months. This infrequency contributed to the recruitment target not being met within study timelines, a common finding in RCTs.[35] The infrequent clinics are representative of the low number of individuals diagnosed with FH in the UK (8%) and worldwide (1%–15%).[36] Ongoing efforts to increase the detection of FH through cascade testing continue to improve these statistics,[36] which should increase the number of eligible families and clinics in a future trial. The study recruited 38% of eligible families, higher than the 34% reported in a similar study conducted in adults with FH[37] and the 22% reported in a family-based study to reduce CVD risk in children without FH.[38] Recruitment could be improved by telephoning non-responders[39] and incorporating a research visit into a lipid clinic appointment to reduce the time burden on participants, a common barrier to participation in this study. Once recruited, the acceptability of the trial was high regardless of randomisation, as reflected by retention of 97%. This study has also demonstrated the feasibility of non-affected parents participating in the trial when the affected parent cannot. As treatment is most beneficial when initiated in childhood,[10] and the family-based delivery of the intervention was integral to the participants' acceptability, non-affected parents should continue to be eligible to facilitate recruitment of children in a future trial.

Participant acceptability and compliance with recording dietary and physical activity behaviours was higher than previous research.[40 41] Battery malfunction due to the age of the accelerometers used accounted for most missing sedentary behaviour outcomes and a future trial should use new or recently refurbished activPALs. The low collection of body composition outcomes can be overcome in a future trial by providing sites with the equipment, and the updated EQ-5D-5L version of the adult QoL questionnaire should be used, which reduces the occurrence of the ceiling effect found in this study.[42]

In contrast to a previous trial conducted in adults with FH,[14] the intervention was implemented with high fidelity, likely due to the comprehensive manuals created. The adaptations made were almost all in response to the

**Table 2** Mean daily intakes of nutrients and clinical outcome measures (SBP, DBP, total cholesterol, LDL-C, HDL-C and HRQoL) at baseline and endpoint assessments for control and intervention groups and a comparison of the differences between groups at the endpoint assessment unadjusted and adjusted for baseline values*

| Outcome | Group | Children | | | | | | | Adults | | | | | | |
|---|---|---|---|---|---|---|---|---|---|---|---|---|---|---|---|
| | | n | Baseline | n | Endpoint | Endpoint difference (95% CI) | n | Adjusted difference (95% CI)† | n | Baseline | n | Endpoint | Endpoint difference (95% CI) | n | Adjusted difference (95% CI)† |
| Fat intake (% TEI) | Intervention | 9 | 33.0 (6.4) | 9 | 28.3 (5.1) | −5.6 (−10.2 to −0.9) | 9 | −5.3 (−8.9 to −1.5) | 8 | 35.4 (7.8) | 8 | 31.2 (5.9) | −0.3 (−6.9 to 6.2) | 8 | −2.3 (−7.4 to 2.9) |
| | Control | 10 | 32.9 (4.8) | 9 | 33.9 (4.1) | | 8 | | 9 | 32.0 (5.7) | 8 | 31.6 (6.3) | | 8 | |
| SFA intake (% TEI) | Intervention | 9 | 12.6 (2.0) | 9 | 10.3 (2.6) | −1.8 (−4.2 to 0.5) | 9 | −1.8 (−4.3 to 0.8) | 8 | 12.1 (3.8) | 8 | 9.9 (2.6) | −1.1 (−4.3 to 2.2) | 8 | −1.9 (−4.3 to 0.5) |
| | Control | 10 | 12.1 (2.9) | 9 | 12.1 (2.1) | | 8 | | 9 | 10.8 (3.4) | 8 | 10.9 (3.3) | | 8 | |
| MUFA intake (% TEI) | Intervention | 9 | 11.6 (4.1) | 9 | 9.4 (2.0) | −2.8 (−4.9 to −0.6) | 9 | −3.2 (−5.3 to −1.01) | 8 | 13.5 (3.7) | 8 | 11.5 (3.4) | 0.9 (−2.8 to 4.6) | 8 | 0.3 (−3.3 to 3.9) |
| | Control | 10 | 11.3 (1.8) | 9 | 12.2 (2.3) | | 8 | | 9 | 12.1 (2.6) | 8 | 10.6 (3.5) | | 8 | |
| PUFA intake (% TEI) | Intervention | 9 | 4.8 (1.4) | 9 | 4.8 (1.1) | −0.4 (−1.7 to 0.8) | 9 | −0.6 (−1.9 to 0.7) | 8 | 5.7 (1.5) | 8 | 5.6 (1.6) | 0.1 (−1.9 to 2.2) | 8 | 0.03 (−2.1 to 2.2) |
| | Control | 10 | 4.6 (1.1) | 9 | 5.3 (1.3) | | 8 | | 9 | 5.2 (1.4) | 8 | 5.5 (2.2) | | 8 | |
| Cholesterol intake (mg) | Intervention | 9 | 181.9 (71.2) | 9 | 149.4 (54.6) | −34.1 (−117.2 to 48.7) | 9 | −24.1 (−100.9 to 52.7) | 8 | 219.2 (84.6) | 8 | 204.4 (30.5) | 63.9 (−39.6 to 167.5) | 8 | 28.1 (−76.6 to 132.8) |
| | Control | 10 | 220.9 (95.2) | 9 | 183.5 (34.7) | | 8 | | 9 | 160.9 (60.1) | 8 | 37.4 (105.8) | | 8 | |
| Fibre intake (g) | Intervention | 9 | 20.0 (5.2) | 9 | 24.2 (6.4) | 7.8 (1.8 to 13.9) | 9 | 5.2 (−0.7 to 10.9) | 8 | 15.5 (5.7) | 8 | 20.6 (7.8) | −2.1 (−10.2 to 6.1) | 8 | 1.1 (−6.7 to 8.8) |
| | Control | 10 | 15.4 (4.0) | 9 | 16.3 (5.7) | | 8 | | 9 | 22.4 (10.6) | 8 | 22.6 (7.4) | | 8 | |
| Fruit and vegetable intake (portions) | Intervention | 9 | 3.2 (1.7) | 9 | 4.2 (0.9) | 2.2 (1.4 to 3.0) | 9 | 2.2 (1.2 to 3.2) | 8 | 3.3 (2.0) | 8 | 4.1 (1.5) | 1.2 (−0.3 to 2.8) | 8 | 1.6 (0.1 to 3.1) |
| | Control | 10 | 2.2 (1.1) | 9 | 2.0 (0.6) | | 8 | | 9 | 4.5 (1.8) | 8 | 2.9 (1.3) | | 8 | |
| SBP (mm Hg) | Intervention | 10 | 104.5 (10.5) | 9 | 104.9 (6.3) | −3.1 (−10.9 to 4.8) | 9 | −1.6 (−8.6 to 5.3) | 8 | 115.5 (15.1) | 8 | 111.9 (3.6) | −8.1 (−21.2 to 4.9) | 8 | −6.8 (−15.9 to 1.4) |
| | Control | 12 | 110.3 (102) | 11 | 107.9 (8.3) | | 11 | | 9 | 117.5 (14.5) | 9 | 120 (4.5) | | 9 | |
| DBP (mm Hg) | Intervention | 10 | 62.9 (6.7) | 9 | 58.9 (4.9) | −4.1 (−10.4 to 2.3) | 9 | −2.2 (−8.1 to 3.7) | 8 | 75.4 (10.1) | 8 | 71.6 (7.2) | −3.2 (−11.4 to 5.1) | 8 | −6.8 (−12.7 to −0.9) |
| | Control | 12 | 68.9 (9.2) | 11 | 63.0 (2.4) | | 11 | | 9 | 69.7 (8.6) | 9 | 74.8 (8.6) | | 9 | |
| Total cholesterol (mmol/L) | Intervention | 8 | 5.2 (1.9) | 8 | 5.0 (1.7) | −0.06 (−1.4 to 1.5) | 8 | −0.3 (−0.9 to 0.4) | 7 | 5.0 (0.9) | 7 | 4.8 (0.3) | −0.08 (−1.3 to 1.2) | 7 | −0.4 (−1.2 to 0.4) |
| | Control | 10 | 4.8 (0.8) | 8 | 4.9 (0.8) | | 8 | | 9 | 4.6 (0.9) | 8 | 4.9 (0.5) | | 8 | |
| LDL-C (mmol/L) | Intervention | 8 | 3.5 (0.8) | 8 | 3.2 (1.6) | −0.2 (−1.5 to 1.1) | 8 | −0.36 (−0.82 to 0.12) | 7 | 3.2 (0.7) | 7 | 2.8 (0.8) | −0.2 (−1.2 to 0.8) | 7 | −0.56 (−1.33 to 0.12) |
| | Control | 10 | 3.2 (0.8) | 8 | 3.3 (0.7) | | 8 | | 9 | 2.7 (0.8) | 8 | 2.9 (1.1) | | 8 | |
| HDL-C (mmol/L) | Intervention | 8 | 1.4 (0.3) | 8 | 1.4 (0.3) | 0.1 (−0.2 to 0.4) | 8 | 0.01 (−0.1 to 0.2) | 7 | 1.47 (0.16) | 7 | 1.39 (0.22) | −0.13 (−0.38 to 0.13) | 8 | −0.17 (−0.38 to 0.04) |
| | Control | 10 | 1.2 (0.3) | 8 | 1.3 (0.1) | | 8 | | 9 | 1.45 (0.27) | 8 | 1.52 (0.09) | | 8 | |
| HRQoL‡ (score 0–100) | Intervention | 10 | 85.1 (9.8) | 9 | 89.0 (7.8) | 16.7 (7.6 to 25.7) | 9 | 5.9 (−0.3 to 12.0) | | | | | | | |
| | Control | 12 | 73.4 (9.9) | 11 | 72.3 (3.3) | | 11 | | | | | | | | |

*All data have been included, the 'n' columns indicate the number of participants included in each estimated group mean value and estimated mean differences between groups.
†Adjusted for baseline values, this difference estimate includes only participants for whom paired baseline and endpoint data were collected. baseline and endpoint data were collected from adult participants.
‡HRQoL data presented for children only due to 'ceiling effect' that occurred in the data collected from adult participants.
DBP, diastolic blood pressure; HDL-C, high-density lipoprotein cholesterol; HRQoL, health-related quality of life; LDL-C, low-density lipoprotein cholesterol; MUFA, monounsaturated fatty acids; PUFA, polyunsaturated fatty acids; SBP, systolic blood pressure; SFA, saturated fatty acids; TEI, total energy intake.

**Table 3** Median (range) values for BMI, body composition, daily MVPA and sedentary time at baseline and endpoint assessments for control and intervention groups and the median change (range) between these timepoints for both groups for children and adults*†

| Outcomes | Group | Children | | | | | | Adults | | | | | |
|---|---|---|---|---|---|---|---|---|---|---|---|---|---|
| | | n | Baseline median (range) | n | Endpoint median (range) | n | Median change (range) | n | Baseline median (range) | n | Endpoint median (range) | n | Median change (range) |
| BMI (z-score children; kg/m² adults) | Intervention | 10 | 0.71 (−1.46 to 3.14) | 9 | 0.54 (−1.74 to 3.18) | 9 | −0.20 (−0.39 to 0.07) | 8 | 24.9 (21.9-29.1) | 8 | 25.5 (22.2-28.9) | 8 | 0.1 (−1.31 to 0.68) |
| | Control | 12 | 0.41 (−1.79 to 2.35) | 11 | 0.31 (−1.72 to 1.45) | 11 | 0.07 (−0.28 to 0.30) | 9 | 24.0 (21.7-28.1) | 9 | 23.2 (22.5-27.5) | 9 | −0.41 (−1.85 to 1.35) |
| Body fat (%) | Intervention | 6 | 21.2 (13-24.5) | 5 | 22.4 (12.9-23.7) | 5 | −0.1 (−1.0 to 0.9) | 4 | 26.4 (23-36.5) | 4 | 26.8 (21.6-33.7) | 4 | −0.6 (−2.8 to 0.5) |
| | Control | 6 | 27.9 (14.4-37) | 5 | 23.8 (16.3-29.2) | 5 | 0.2 (−1.4 to 0.3) | 4 | 29.2 (22.4-33.2) | 4 | 26.9 (22.7-32.5) | 4 | −0.9 (−3.5 to 0.3) |
| Fat-free mass (kg) | Intervention | 6 | 41.3 (25.4-56.9) | 5 | 43.1 (29.4-56.2) | 5 | −0.6 (−1.0 to −0.3) | 4 | 45.1 (41.8-52.9) | 4 | 44.9 (40.8-54.4) | 4 | −0.2 (−1 to 1.5) |
| | Control | 6 | 38.2 (27.2-43.8) | 5 | 39.1 (29.2-47.3) | 5 | 1.6 (0.3-3.5) | 4 | 52.4 (40.7-58.2) | 4 | 51.8 (40.6-58) | 4 | −0.2 (−2.2 to 0.9) |
| Daily MVPA (min) | Intervention | 10 | 67.1 (59.9-84.6) | 10 | 62.2 (42.2-86.2) | 10 | −4.2 (−14.3 to 2.4) | 8 | 37.7 (28.0-63.1) | 8 | 47.4 (27.9-79.9) | 8 | 4.3 (1.2-17.2) |
| | Control | 10 | 51.5 (41.9-71.9) | 8 | 45.9 (38.7-59.4) | 6 | −5.9 (−19.1 to 12.4) | 8 | 34.5 (21.4-59.0) | 8 | 41.9 (24.7-65.9) | 7 | 6 (−22.2 to 19.1) |
| Daily sedentary time (min) | Intervention | 9 | 551 (497-553) | 8 | 560 (526-608) | 7 | 14 (−41 to 105) | 8 | 593 (543-624) | 6 | 552 (529-583) | 6 | −34 (−52 to −7) |
| | Control | 10 | 515 (494-633) | 6 | 595 (532-674) | 5 | 81 (10-160) | 8 | 571 (478-619) | 8 | 537 (293-679) | 5 | 3.8 (−32 to 19) |

*All data have been included, the 'n' columns indicate the number of participants included in each estimated group median value and estimated median within-group changes from baseline to endpoint.
†Ranges are presented due to small sample sizes.
BMI, body mass index; MVPA, moderate and vigorous physical activity.

individual needs of families. If adaptations do not undermine the mechanisms of action by which the intervention is proposed to have its effect, tailoring interventions to suit different contexts can have beneficial effects on the outcomes.[43] As the intervention was systematically developed,[18] the effect of adaptations can be evaluated in a definitive trial.[21]

Participants' dietary intakes fell short of treatment recommendations at baseline, as reported by others previously.[13 44] This was despite all adults having previously received dietetic advice on diagnosis, suggesting the specifically designed behaviour change intervention has advantages over routine advice provided in outpatient lipid clinics. The qualitative findings serve to demonstrate that the intervention was highly acceptable to families with FH, who were able and willing to change their lifestyle behaviours. This was largely driven by the family-based approach, also adopted by a previous non-randomised trial, which successfully improved dietary behaviours of children with FH.[44] However, only the behaviours of the children were targeted and eight dietetic sessions were provided.[44] While not powered to detect changes, the findings suggest that the pragmatic intervention evaluated in the current study may achieve dietary improvements in both adults and children without the need for multiple dietetic sessions.

Unlike previous research,[14] the results suggest physical activity and sedentary behaviours of adults with FH may be improved by the intervention developed for the current study. However, trends towards improvement among adults in the control group were also observed. This finding may be explained by the *reactivity bias* associated with wearing accelerometers, in which participants modify their habitual physical activity in response to wearing accelerometers.[45] In contrast, the results of this study do not suggest that physical activity or sedentary behaviours were improved among children in either group. Participants received feedback about the disparities between their dietary intakes and the recommendations, but not for physical activity. Parents of inactive children typically overestimate the time they spend in physical activity, as do the children themselves.[46] Given the low rates of children meeting the physical activity recommendations in this study, feedback on current physical activity levels in comparison to the recommendations should be provided in a future trial to increase motivation to change this behaviour.

### Strengths and limitations

The participants were highly educated and mostly white Europeans, limiting the generalisability of the findings to all families with FH, a common condition affecting all ethnicities.[2] The qualitative sample did not include any participants of different ethnicities and it cannot be concluded that data saturation was achieved, limiting the validity of the qualitative findings.[47] The 12-week follow-up period in this study does not enable exploration of longer term effects, which are of importance to

treatment for a lifelong condition like FH.[11] Individuals attend annual lipid clinic appointments and it would be feasible to extend the follow-up period to 12 months in a future trial. The dietary intake estimates in this study are limited by the misreporting bias associated with all self-report methods of dietary assessment.[48] A future trial should include objective measures of dietary intakes to validate the estimates of the nutrients targeted by the intervention, such as blood concentrations of fatty acids and phytosterols.[49] Adherence to pharmacological treatment influences LDL-C concentrations and this was not formally assessed in this study. The validated Morisky Eight-item Medication Adherence Scale is suggested for a future trial.[50]

## CONCLUSION

This study has addressed the main uncertainties ahead of a definitive trial, and if the identified refinements are made to the study design then a fully powered RCT is feasible and warranted. Such a trial will address the unanswered question of the potential benefits of the current diet and physical activity treatment recommendations on LDL-C treatment goals and CVD risk in children and adults with FH.

**Author affiliations**
[1]National Institute for Health Research Bristol Biomedical Research Centre (NIHR Bristol BRC), University Hospitals Bristol and Weston NHS Foundation Trust, Bristol, UK
[2]University of Bristol, Bristol, UK
[3]Bristol Medical School, University of Bristol, Bristol, UK
[4]Department of Clinical Biochemistry, University Hospitals Bristol and Weston NHS Foundation Trust, Bristol, UK
[5]St George's University Hospitals NHS Foundation Trust, London, UK
[6]National Centre for Sport and Exercise Medicine, School of Sport, Exercise and Health Sciences, Loughborough University, Loughborough, UK
[7]National Institute for Health Research (NIHR) Leicester Biomedical Research Centre (BRC), University Hospitals of Leicester NHS Trust, Leicester, UK

**Acknowledgements** We would like to thank Dr Coxson, Dr Edmonds and all members of the research teams at University Hospitals Bristol and Weston, St George's University Hospitals and Royal United Hospitals Bath NHS Foundation Trusts. DJS and AET acknowledge support from the National Institute for Health Research (NIHR) Leicester Biomedical Research Centre.

**Contributors** FJK, JPHS, FEL, GB, CW and DJS contributed to the design of the study. FJK and AS analysed and interpreted the qualitative data. FJK, AET, FEL and JPHS analysed and interpreted the quantitative data. FJK and LH devised and conducted the statistical analysis plan for this analysis. FJK prepared the manuscript, which was critically revised by all coauthors. All authors approved the final version.

**Funding** This study was funded by the NIHR Biomedical Research Centre at University Hospitals Bristol and Weston NHS Foundation Trust and the University of Bristol.

**Competing interests** None declared.

**Patient consent for publication** Not required.

**Ethics approval** The trial received ethical approval from the Cornwall and Plymouth Research Ethics Committee (reference: 18/SW/0121).

**Provenance and peer review** Not commissioned; externally peer reviewed.

**Data availability statement** Data are available upon reasonable request. Data from this study are available on request.

**ORCID iDs**
Fiona Jane Kinnear http://orcid.org/0000-0002-4090-1554
Aidan Searle http://orcid.org/0000-0001-9860-3253
David J Stensel http://orcid.org/0000-0001-9119-8590

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
