## [Reviewer comments · BMJ Open]

ARTICLE DETAILS

TITLE (PROVISIONAL)	Reducing cardiovascular disease risk amongst families with familial hypercholesterolaemia by improving diet and physical activity: A randomised controlled feasibility trial
AUTHORS	Kinnear, Fiona; Lithander, Fiona; Searle, Aidan; Bayly, Graham; Wei, Christina; Stensel, David; Thackray, Alice; Hunt, Linda; Shield, Julian

VERSION 1 – REVIEW

REVIEWER	Gerald F Watts School of Medicine University of Western Australia
REVIEW RETURNED	19-Sep-2020

GENERAL COMMENTS	An excellent feasibility trial showing that diet and physical activity may be improved in families with FH sufficient to justify a formal RCT. Strengths are relevance and developmental work with the lifestyle intervention; clear evidence of acceptability, recruitment, retention and efficacy. The molecularly defined diagnosis of FH is a strength. Weaknesses are short period of intervention, lack of multiethnicity and non-selection of subjects for having abnormal health behaviours. The authors should revise and include an estimated sample size based on the efficacy data for the formal RCT. How are the outcome variables adjusted for family clustering of effects in these types of trials ? Do the findings have implications for managing CV risk attributed to other inherited cause of premature CAD in families ?
---

REVIEWER	Krzysztof Chlebus National Center for FH Poland 1-st Dept. of Cardiology Medical University of Gdansk Poland
REVIEW RETURNED	15-Oct-2020

GENERAL COMMENTS	Valuable and well-planned work. Its field is tough and neglected for commercial reasons, but important and necessary from a public health perspective. Its disadvantage is the short observation time and a small cohort of patients, but it provides evidence for the creation of a future adequately powered trial.
---

VERSION 1 – AUTHOR RESPONSE

Reviewer 1 comments:

An excellent feasibility trial showing that diet and physical activity may be improved in families with FH sufficient to justify a formal RCT. Strengths are relevance and developmental work with the lifestyle intervention; clear evidence of acceptability, recruitment, retention and efficacy. The molecularly defined diagnosis of FH is a strength.

Weaknesses are short period of intervention, lack of multi ethnicity and non-selection of subjects for having abnormal health behaviours. The authors should revise and include an estimated sample size based on the efficacy data for the formal RCT. How are the outcome variables adjusted for family clustering of effects in these types of trials? Do the findings have implications for managing CV risk attributed to other inherited cause of premature CAD in families?

Response from Authors to the comments received from Reviewer 1:

Thank you for the positive comments regarding our manuscript.

With regard to sample size for a full trial, we have already included a sample size estimation for a fully randomised trial with a single child within each family (Page 11). We do agree that if more than one child were to be recruited within families, account would need to be taken of intra-family correlation of outcomes. We have added a sentence to this effect in the text. (Page 11, lines 297-301)

'If multiple affected children from a family were to be recruited to any following trial, an estimate of the intra-class correlation would be needed to calculate an inflation factor for the sample size.

Furthermore, mixed models may be needed in the final analyses. As only one family in the present trial included more than one affected child, it was not possible to estimate the intra-class correlation.'

Given that the intervention was developed specifically for families with FH, we do not believe that it would be appropriate to extrapolate the findings to the management of CV risk amongst individuals with other inherited causes of premature CAD. However, the approach we took could certainly be replicated to determine the feasibility, acceptability, and potential efficacy of delivering the intervention to families of similar population groups. We have not included this in the text as we do not feel this was the central thrust of our work which was only related to FH.

Comments received from Reviewer 2

Valuable and well-planned work. Its field is tough and neglected for commercial reasons, but important and necessary from a public health perspective. Its disadvantage is the short observation time and a small cohort of patients, but it provides evidence for the creation of a future adequately powered trial.

Response from the Authors to comments received from Reviewer 2:

Thank you for your positive comments regarding the manuscript and your recognition of the value of the research undertaken.